# Effects of an Acute Pilates Program under Hypoxic Conditions on Vascular Endothelial Function in Pilates Participants: A Randomized Crossover Trial

**DOI:** 10.3390/ijerph17072584

**Published:** 2020-04-09

**Authors:** Kyounghwa Jung, Jongbeom Seo, Won-Sang Jung, Jisu Kim, Hun-Young Park, Kiwon Lim

**Affiliations:** 1Department of Physical Education, Konkuk University, 120 Neungdong-ro, Gwangjin-gu, Seoul 05029, Korea; pilateslab@konkuk.ac.kr (K.J.); syk1528@konkuk.ac.kr (J.S.); 2Physical Activity and Performance Institute (PAPI), Konkuk University, 120 Neungdong-ro, Gwangjin-gu, Seoul 05029, Korea; jws1197@konkuk.ac.kr (W.-S.J.); kimpro@konkuk.ac.kr (J.K.); parkhy1980@konkuk.ac.kr (H.-Y.P.); 3Department of Sports Medicine and Science, Graduate School, Konkuk University, 120 Neungdong-ro, Gwangjin-gu, Seoul 05029, Korea

**Keywords:** Pilates exercise, hypoxia, metabolic parameters, cardiac function, endothelial function

## Abstract

This study aimed to compare the effects of an acute Pilates program under hypoxic vs. normoxic conditions on the metabolic, cardiac, and vascular functions of the participants. Ten healthy female Pilates experts completed a 50-min tubing Pilates program under normoxic conditions (N trial) and under 3000 m (inspired oxygen fraction = 14.5%) hypobaric hypoxia conditions (H trial) after a 30-min exposure in the respective environments on different days. Blood pressure, branchial ankle pulse wave velocity, and flow-mediated dilation (FMD) in the branchial artery were measured before and after the exercise. Metabolic parameters and cardiac function were assessed every minute during the exercise. Both trials showed a significant increase in FMD; however, the increase in FMD was significantly higher after the H trial than that after the N trial. Furthermore, FMD before exercise was significantly higher in the H trial than in the N trial. In terms of metabolic parameters, minute ventilation, carbon dioxide excretion, respiratory exchange ratio, and carbohydrate oxidation were significantly higher but fat oxidation was lower during the H trial than during the N trial. In terms of cardiac function, heart rate was significantly increased during the H trial than during the N trial. Our results suggested that, compared to that under normoxic conditions, Pilates exercise under hypoxic conditions led to greater metabolic and cardiac responses and also elicited an additive effect on vascular endothelial function.

## 1. Introduction

Pilates is a series of around 500 exercises inspired by calisthenics, yoga, and ballet. Pilates is a system of exercises that emphasizes control, breathing, flow, accuracy, centering, and concentration, by which the person perceives the body and activates the muscles [1]. In particular, Pilates programs using tubing bands have the advantage of minimizing the possibility of injury to joints and muscles and freely adjusting the intensity according to one’s own muscle strength and physical fitness [2,3]. Pilates programs have relatively low intensity compared to endurance exercises that can attract the attention of sedentary/overweight/obese people; in addition, Pilates also strengthen the core and increase the flexibility of the body [4,5].

However, Pilates programs have been proposed to have low positive impact on cardiopulmonary and vascular functions due to the low exercise intensity [6,7]. Therefore, previous studies have employed a combined approach composed of Pilates and endurance exercises to induce development of the neuromuscular system and improvement of the cardiovascular system [8,9,10,11]. However, endurance exercise enhances joint load and mechanical work, which, in turn, increases the risk of damage to the musculoskeletal system of sedentary/overweight/obese people [12,13,14].

Nowadays, hypoxic conditions are utilized to achieve a greater metabolic effect while lowering the mechanical loads during physical exercise [14,15]. The hypoxic conditions allow the achievement of a higher metabolic demand. Moreover, lower exercise intensity tends to be more protective of the muscle/joints of sedentary/overweight/obese people [14,16,17]. Consequently, employment of sustained hypoxic conditions might be beneficial for weight management programs.

Hypoxic conditions elicit a reduction in arterial stiffness and increase in the blood flow within the skeletal muscle vascular beds [13,18,19]. Intermittent hypoxic exposure enhances endothelial-derived nitric oxide-mediated mechanisms that are primarily involved in the vasodilation of muscular arteries; as a result, it is frequently applied to lower the blood pressure of patients with hypertension [20,21]. In addition, exercise intervention under hypoxic conditions induces an additive effect on the secretion of various vasodilators (e.g., adenosine, prostaglandin, and nitric oxide [NO]) and vascular endothelial growth factor, which leads to an additive effect on the improvement of vasodilation, reduction of arterial stiffness, and microcirculation through improved hemorheological functions [13,15,22,23,24,25].

As mentioned earlier, owing to its lower intensity, Pilates program is not effective in enhancing cardiovascular function. Therefore, it is crucial to examine whether a combination of Pilates and hypoxic condition exerts additive effects on cardiac and vascular functions. The combination of Pilates program and hypoxic conditions might be an effective approach to improve musculoskeletal function and cardiovascular function while reducing the risk of injury by reducing the exercise load and mechanical stress. Although the exercise type is different, many previous studies have reported that exercise intervention under hypoxic conditions led to a greater improvement in the body composition, metabolic function, and cardiovascular function with lower exercise intensity than that induced by exercise intervention under normoxic conditions [12,15,17,24,26,27]. Therefore, it is crucial to examine the combined effects of Pilates programs and hypoxic conditions in the improvement of metabolic, cardiac, and vascular functions. 

However, the present study presents a new insight into combination of Pilates program and hypoxic conditions by examining the differences in the metabolic, cardiac, and vascular functions responses under hypoxic and normoxic conditions in Pilates participants who can perform Pilates programs with accurate movements. It is necessary to examine these effects, before employing this combined approach for sedentary/overweight/obese people.

Therefore, we conducted a randomized crossover trial to compare the effects of an acute Pilates program under hypoxic conditions vs. normoxic conditions on metabolic, cardiac, and vascular function in experienced Pilates participants.

## 2. Materials and Methods 

### 2.1. Subjects

The study subjects included 10 healthy women who were experienced in Pilates (age, 26.4 ± 3.0 years; height, 162.2 ± 4.1 cm; weight, 50.8 ± 5.8 kg; body mass index, 19.3 ± 1.6; fat-free mass, 38.2 ± 3.8 kg; percent of body fat, 24.6% ± 5.0%), were non-smokers, and had no history of musculoskeletal, cardiovascular, or pulmonary disease. The consolidated standards of reporting trial (CONSORT) flow diagram is shown in Figure 1. The participants received information about the purpose and process of this study and provided informed consent prior to the start of the study. The characteristics of the participants are presented in Table 1. The trial was registered and disclosed to the Institutional Review Board of Konkuk University (7001355-201909-HR-334) and clinical trial information was registered in the Clinical Research Information Service (KCT0004517) in Korea and was conducted according to the guidelines of the declaration of Helsinki.

### 2.2. Study Design

The study design is shown in Figure 2. All subjects visited the laboratory three times during the experimental period. During the first visit, they underwent a familiarization trial under normoxic conditions (inspired oxygen fraction; F_i_O_2_ = 20.9%) prior to the main trial for adaptation to the Pilates program. On the second and third occasions, the subjects underwent a randomized crossover experimental trial under either hypoxic conditions (F_i_O_2_ = 14.5%, a 3000 m simulated altitude; H trial) or normoxic conditions (N trial). All subjects had a washout period of at least 7 days between each trial. The order of the conditions for each experimental trial was randomized, and each subject underwent the protocol at the same time at each visit. However, the subjects did not undergo blinded experiments under environmental conditions.

During all trials, all subjects were exposed to each environmental condition for 30 min, and then, a tubing Pilates program was performed for 50 min under each environmental condition. The Pilates program using tubing was composed of 25 types of motion, each performed for 2 min. The Pilates motion was configured as follows: roll up and down, biceps, arm circles, teaser, rolling like a ball, spine twist and arm extension, tubing mermaid, cobra, swimming, double kicks and arm circles, swan, cat, thigh stretching, hug a tree, squat, row, saw, hip pull, the hundred, lats pull three way, leg arc, scissor, helicopter, right side leg pull, and left side leg pull [28].

Metabolic function (minute ventilation: VE, oxygen uptake: VO_2_, carbon dioxide excretion: VCO_2_, respiratory exchange ratio: RER, carbohydrate oxidation: CHO, fatty acid oxidation: FAO, and energy expenditure: EE) and cardiac function (heart rate: HR, stroke volume: SV, cardiac output: CO, end-diastolic volume: EDV, end-systolic volume: ESV, and ejection fraction: EF) were measured every minute during the Pilates program. The sum of the values measured for 50 min was used as the result (with the exception of RER and EF, for which average values were used). Blood pressure (BP), branchial ankle pulse wave velocity (baPWV), and flow-mediated dilation (FMD) in the branchial artery were measured as indicators of vascular function before and after the Pilates program. 

In all trials, the Pilates program was performed in a 9 m (width) × 7 m (length) × 3 m (high) chamber at a temperature of 23 ± 1 °C and humidity of 50% ± 5% regulated by an environmental control chamber (NCTC-1, Nara Control, Seoul, Korea).

### 2.3. Body Composition

Height, weight, body mass index, fat-free mass, and percent body fat were analyzed as the body composition parameters using bioelectrical impedance analysis equipment (Inbody 770, Inbody, Seoul, Korea). All subjects fasted overnight prior to the body composition measurements. The subjects wore lightweight clothing and were asked to remove any metal items.

### 2.4. Metabolic Function

All metabolic function parameters (e.g., VE, VO_2_, VCO_2_, RER, CHO, FAO, and EE) were measured every minute. VE, VO_2_, VCO_2_, and RER were measured using the K5 auto metabolism analyzer (Cosmed, Rome, Italy) and a breathing valve in the form of a facemask during the 50-min Pilates program. The measured rates of whole body VO_2_ and VCO_2_ were used to estimate the rates of EE (kcal/min) and the disappearance rates (g/min) via CHO and FAO substrates. The following formulae were used to estimate CHO, FAT, and EE during 50-min Pilates program: CHO = 4.210 × VCO_2_ − 2.962 × VO_2_, FAT = 1.695 × VO_2_ − 1.701 × VCO_2_, EE = 4.07 × CHO + 9.75 × FAT [29]. 

The sum of the values measured during the 50-min program was used as the result (with the exception of RER, for which the average value was used).

### 2.5. Cardiac Function

The cardiac function parameters, HR, SV, CO, EDV, ESV, and EF, were assessed noninvasively every minute using a thoracic bioelectrical impedance device (PhysioFlow PF-05, Paris, France) during the 50-min Pilates program. The sum of the values measured during the 50-min program was used as the result (with the exception of EF, for which the average value was used).

### 2.6. Vascular Function

BP, baPWV, and FMD in the branchial artery were measured as indicators of vascular function before and after the Pilates program. BP was measured twice using an autonomic BP monitor (HBP-9020, Omron, Tokyo, Japan) before and after the Pilates program, and the average value was used as the result. Resting BP was measured after 30 min of exposure to each environmental condition, and post-exercise BP was measured immediately following the Pilates program for each environmental condition.

Pulse wave velocity is the best available parameter for assessing arterial stiffness. Therefore, baPWV was measured using an automatic oscillometric device (VP-1000plus, Omron, Osaka, Japan) before and after the Pilates program. Resting baPWV was measured after 30 min of exposure to each environmental condition, and post-exercise baPWV was measured within 10 min after completion of the Pilates program for each environmental condition.

FMD refers to dilation (widening) of an artery when blood flow increases in that artery. The primary cause of FMD is the release of NO by endothelial cells. Therefore, FMD of the branchial artery was measured to evaluate the vascular endothelial function using noninvasive Doppler ultrasound (UNEX-EF, Tokyo, Japan) before and after the Pilates program. The resting FMD was measured after 30 min of exposure to each environmental condition. After fixing the ultrasound equipment to the brachial artery region 3–5 cm above the elbow, the diameter of the medial muscular artery was measured by Doppler. After the measurement, blood was removed for 5 min by increasing 50 mmHg based on resting BP. After 5 min, deflation was automatically recorded for next 2 min to evaluate the diameter and blood flow rate, and the calculated values of FMD (FMD = (reactive hyperemia diameter - baseline diameter) × 100%) were used. Post-exercise FMD was measured in the same way within 20 min after completion of the Pilates program for each environmental condition.

### 2.7. Statistical Analysis

All statistical analyses were conducted using the SPSS software version 25.0 for Windows (IBM Corp., Armonk, NY, USA). Data are presented as means ± standard deviations. The assumption of normality and homoscedasticity was verified using the Shapiro-Wilks W-test prior to the parametric tests. First, paired *t*-test was used to compare the metabolic function and cardiac function during the N trial vs. those during the H trial. Second, two-way analysis of variance (ANOVA) with repeated measures was used to assess the presence of interactions (trial × time) in vascular function before and after the Pilates program under each environmental condition. When ANOVA revealed a significant interaction, the Bonferroni test was performed as a post-hoc analysis to identify the differences. We used Cohen’s d where the term effect size reflects the value of a statistic calculated from a sample of data and standardized mean differences. A significance level of *p* < 0.05 was used to determine statistical difference for the mean of effect size and the confidence interval (CI) was reflected at a confidence level of 95%.

## 3. Results

### 3.1. Metabolic Function

As shown in Table 1, a significant difference was found in the values of VE, VCO_2_, RER, CHO, and FAO during the 50-min Pilates program between the two trials. The H trial showed a greater increase in VE (Cohen’s *d*, 0.7; 95% CI, −0.2, 1.6 L/50 min; *p* = 0.011), VCO_2_ (Cohen’s *d*, 0.8; 95% CI, −0.1, 1.7 mL/50 min; *p* = 0.001), RER (Cohen’s *d*, 3.2; 95% CI, 1.8, 4.3; *p* < 0.001), and CHO (Cohen’s *d*, 2.8; 95% CI, 1.5, 3.8 g/50 min; *p* < 0.001) than the N trial. The H trial showed a greater decrease in FAO (Cohen’s *d*, −2.1; 95% CI, −3.1, −1.0 g/50 min; *p* < 0.001) than the N trial. However, there was no significant difference in the values of VO_2_ and EE between the H trial and N trial.

### 3.2. Cardiac Function

Table 2 demonstrates a significant difference in HR during the 50-min Pilates program between the H trial and N trial. The H trial showed a greater increase in HR (Cohen’s *d*, 0.9; 95% CI, 0.0, 1.7 bpm; *p* = 0.011) than the N trial. The values of other cardiac function parameters (e.g., SV, CO, EDV, ESV, and EF) showed no significant differences between the H trial and N trial.

### 3.3. Vascular Function

As shown in Table 3, there was no significant interaction among SBP, DBP, MAP, and PP. Figure 3 shows the changes in baPWV and FMD during the Pilates program in H trial and N trial, and no significant interaction was observed in baPWV. However, a significant interaction was observed between the H trial and N trial with respect to FMD (*p* < 0.01, *η*^2^ = 0.692). Post-hoc analyses found significant improvement in the values of FMD (H trial: Cohen’s *d*, 2.8; 95% CI, 1.5, 3.8; *p* < 0.001; N trial: Cohen’s *d*, 1.6; 95% CI, 0.6, 2.5; *p* < 0.001) during the Pilates program in both trials. In addition, the H trial showed a greater increase in FMD after the Pilates program than the N trial (before Pilates program: Cohen’s *d*, 0.6; 95% CI, −0.3, 1.5; *p* = 0.001; after Pilates program: Cohen’s *d*, 1.6; 95% CI, 0.6, 2.5; *p* < 0.001).

## 4. Discussion

Hypoxic conditions reduce arterial oxygen saturation, which is more affected during exercise due to increased pulmonary blood flow, which limits gas exchange at the alveoli [30]. Exercise under hypoxic conditions results in higher VE and HR than similar workloads performed under normoxic conditions; this finding might be attributed to the need to maintain O_2_ perfusion [31]. However, these physiological adjustments are limited by the decrease in VO_2_max at high altitudes [32]. The absolute rate of CHO and its relative contribution to the fuel mixture at constant load exercise enhance under hypoxic conditions compared to those under normoxic conditions [29,33]. Replacement of the oxidation substrate from fatty acids to glucose at constant load exercise helps to maintain energy supply under hypoxic conditions because of greater energy yield per liter of O_2_ consumed (carbohydrate = 5.05 kcal, fatty acid = 4.69 kcal) when glucose is fully oxidized compared to complete oxidation of fatty acids [34]. This shift in fuel selection indicates a higher relative exercise intensity under hypoxic conditions than that under normoxic conditions when the same absolute workload is performed [31,32]. Therefore, acute exercise under hypoxic conditions elicits an additional metabolic stress that the body must overcome to maintain energy supply within exercising muscles [29].

We examined the effect of an acute 50-min Pilates program under hypoxic conditions on metabolic and cardiac functions of Pilates participants. We observed that the acute Pilates program under hypoxic conditions led to greater increase in VE, VCO_2_, RER, CHO, and HR and decreased FAO during Pilates program compared to that under normoxic conditions. However, no significant difference between the values of VO_2_, SV, CO, EDV, ESV, and EF were observed during the Pilates program performed under hypoxic conditions and normoxic conditions. Moon et al. [31] evaluated the effects of various acute hypoxic conditions (F_i_O_2_ = 16.5%, 14,5%, 12.8%, and 11.2%) vs. normoxic conditions (F_i_O_2_ = 20.9%) on metabolic parameters and cardiac function during constant load (116.7 ± 20.1 W and 60 rpm, 70% HRmax under normoxic conditions) submaximal bicycle exercise. They reported that acute exercise under hypoxic conditions (F_i_O_2_ = 14.5% and below) might be associated with decreases in EDV and ESV and increases in blood lactate level, VE, HR, EF, and CO during submaximal exercise. These changes are an acute compensation response to reduced aerobic exercise capacity due to decreased oxygen delivery and utilization capacity under hypoxic conditions.

Corroborating the results of Moon et al. [31], we observed no significant difference in VO_2_ between hypoxic and normoxic conditions during submaximal exercise; this is likely because the exercise intensity was fixed at a constant level across simulated environment conditions, resulting in the expenditure of same amount of energy. In addition, Hill et al. [35] and Mazzeo [36] reported that the lack of difference in VO_2_ during submaximal exercise between both environmental conditions is due to an increase in VE, HR, and CO because of acute hypoxic conditions. Furthermore, exercise under hypoxic conditions results in a decrease in the partial pressure of oxygen in the arterial blood, which leads to metabolic acidosis due to increase in ATP synthesis by the anaerobic metabolic process, increase in hydrogen ion, and decrease in pH [37]. Ramos-Campo et al. [38] investigated the effects of various hypoxic conditions on metabolic and acid-base balance, blood oxygenation, electrolyte, and half-squat performance parameters during high-resistance circuit (HRC) training. They reported that various normobaric hypoxic conditions during HRC exercise reduced blood oxygenation, pH, and HCO_3_^−^ and increased blood lactate level, which ultimately decreased muscular performance. Consequently, metabolic changes in ATP synthesis led to a shift in fuel selection, resulting in increased carbohydrate oxidation. Indeed, Peronnet et al. [33] demonstrated increased carbohydrate oxidation during exercise under hypoxic conditions, which was consistent with our findings. To compensate for reduced oxygen delivery and utilization capacity under hypoxic conditions, HR was found to be increased by the activation of the sympathetic nervous system through stimulating catecholamine release in the adrenal medulla [39]. Mazzeo [36] stated that the stimulation of β-adrenaline receptors is a major factor responsible for the increase in HR during exercise under hypoxic conditions. We also confirmed greater HR increases during the Pilates program under hypoxic conditions than under normoxic conditions. Overall, our findings on metabolic and cardiac functions were consistent with those of previous studies [36,39]. 

However, our results did not show any differences in SV, EDV, ESV, and CO during the 50-min Pilates program between the two trials. With respect to the cardiac function, EDV and ESV are major determinants of SV, and an increase in EDV and decrease in ESV result in an increase in ventricular contractility and SV [40]. Moon et al. [31] reported that there was no significant difference between EDV values under the hypoxic and normoxic conditions and that ESV decreased to a greater degree under severe hypoxic conditions (FiO_2_ = 12.8% and FiO_2_ = 11.2%) than under normoxic conditions. They showed that these results were probably because of the activation of the sympathetic nervous system and alteration in the catecholamine level of the blood, which was triggered by increased levels of calcium ions in cardiac muscle cells during exercise under severe hypoxic conditions [41]. However, no significant difference were observed between the values of SV, EDV and ESV under moderate hypoxic conditions (FiO_2_ = 16.5%) and those under normoxic conditions; these results were consistent with our findings. However, unlike previous studies, our study did not show an increase in CO even though HR increased to a greater degree during constant load exercise under hypoxic conditions than under normoxic conditions [31]. Therefore, EF, which is a factor determined by EDV and SV, did not change. Naejie [42] showed that the increase in CO under hypoxic conditions was an acute compensation response for a decrease in oxygen diffusion, delivery, and utilization capacity in the lungs and muscle tissues. As mentioned above, our study showed no change in CO, which might be due to the use of the exercise type other than the Pilates program and relatively lighter exercise intensity.

Exercise induces augmented blood flow and subsequently increases vasodilation and upregulates endothelial NO synthase [43,44]. The release of various vasodilators (e.g., adenosine, prostaglandin, and NO) contributes to the mechanism responsible for the vasodilation during exercise [45,46]. Furthermore, hypoxic conditions induce a reduction in arterial stiffness, vasodilation in conduit arteries, and increases in blood flow within the skeletal muscle vascular beds [13,18,19]. The augmented vasodilation during exercise under hypoxic conditions occurs due to the direct release of NO from endothelial sources [47,48]. Casey et al. [48] also showed that vasodilation occurs during exercise under hypoxic conditions due to the NO released from the endothelium. Muscle activation during submaximal exercise with the same relative exercise intensity under hypoxic conditions was higher than that under normoxic conditions [49], and increased muscle activation under hypoxic conditions is likely to increase NO production in the endothelium, and improve vasodilation and endothelial functions [48,50]. Consequently, exercise under hypoxic conditions might elicit upregulation of NO synthase and induce an additive effect on the secretion of various vasodilators (e.g., adenosine, prostaglandin, and NO) and vascular endothelial growth factor, resulting in an additive effect on the improvement of endothelium-dependent vasodilation, reduction of arterial stiffness, and microcirculation through improved hemorheological functions [13,15,22,23,24,25,44].

The present study demonstrated improved branchial artery vasodilation and higher FMD following the Pilates program under hypoxic conditions compared to the Pilates program under normoxic conditions; however, there was no significant difference in BP and baPWV. These results indicated that an acute Pilates program, with light exercise intensity, under hypoxic conditions enhanced endothelium-dependent vasodilation but did not reduce the arterial stiffness in Pilates participants with clinically normal blood pressure and baPWV. Katayama et al. [44] investigated the effect of acute exercise (30% peak oxygen uptake) under hypoxic (FiO_2_ = 12%) vs. normoxic conditions (FiO_2_ = 21%) on FMD. They reported that the %FMD and normalized FMD were significantly higher following exercise under hypoxic conditions (5, 30, and 60-min post-exercise) than under normoxic conditions. These findings suggested that exercise with light intensity under hypoxic conditions had a significant impact on endothelial-mediated vasodilation. The effects of acute exercise under hypoxic conditions on blood pressure and arterial stiffness need to be further validated in future studies while considering subject characteristics (e.g., sedentary/overweight/ obese people), exercise type, and exercise intensity.

In summary, an acute Pilates program under hypoxic conditions leads to a greater metabolic and cardiac response and also elicits an additive effect on vascular endothelial function in the participants compared to the Pilates program under normoxic conditions.

## 5. Limitation of the Study

There are some limitations of the current study. Although our study is the first to combine a Pilates program and hypoxic conditions within a randomized crossover design, the small sample sizes limit our ability to efficiently compare the effects of an acute Pilates program under hypoxic conditions vs. normoxic conditions on metabolic, cardiac, and vascular functions in experienced Pilates participants. Thus, a higher number of subjects may be needed in future studies to accurately assess the efficacy of this combined technique in clinical practice. In addition, the subjects’ dietary habits and physical activities were not regulated.

## 6. Conclusions

The present study is the first to combine a Pilates program and hypoxic conditions. We performed a randomized crossover trial to compare the effects of a Pilates program under hypoxic vs. normoxic conditions on metabolic, cardiac, and vascular functions in Pilates professionals. Consequently, our study confirmed that a Pilates program under a moderately hypoxic condition led to a greater metabolic response by relative higher exercise intensity (e.g., HR). In addition, a Pilates program under hypoxic conditions led to an additive effect on vascular endothelial function compared to that under normoxic conditions in experienced Pilates participants.

## Figures and Tables

**Figure 1 ijerph-17-02584-f001:**
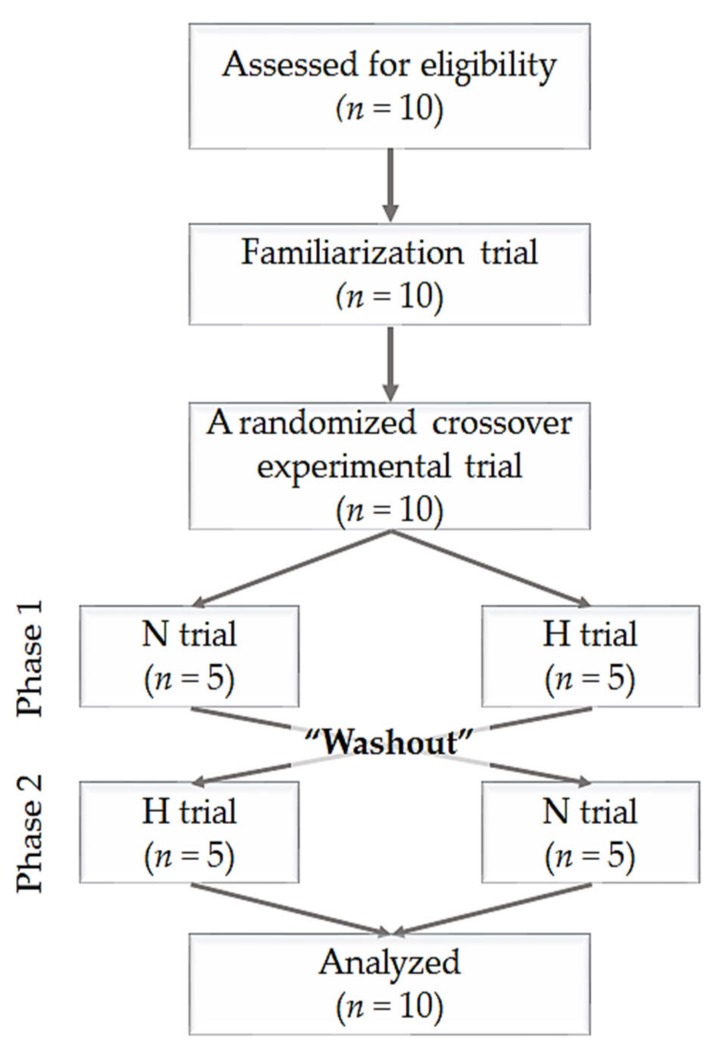
The flow diagram of the consolidated standards of reporting trial. N trial: Pilates program under normoxic conditions, H trial: Pilates program under hypoxic conditions.

**Figure 2 ijerph-17-02584-f002:**
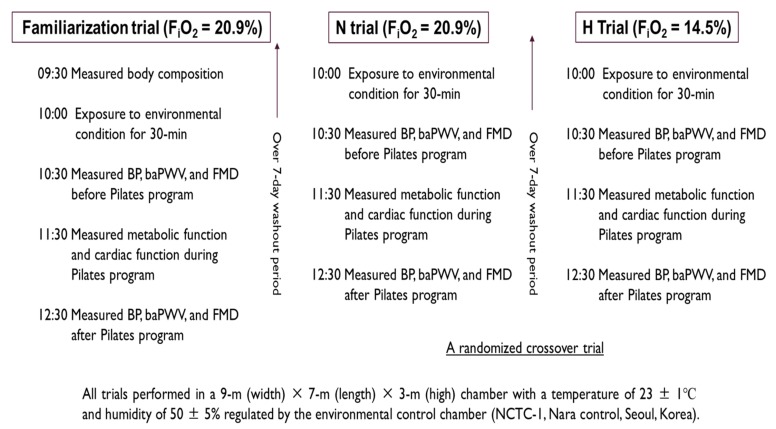
Study design. N trial: Pilates program under normoxic conditions, H trial: Pilates program under hypoxic conditions, FiO_2_: inspired oxygen fraction, BP: blood pressure, baPWV: branchial ankle pulse wave velocity, and FMD: flow-mediated dilation.

**Figure 3 ijerph-17-02584-f003:**
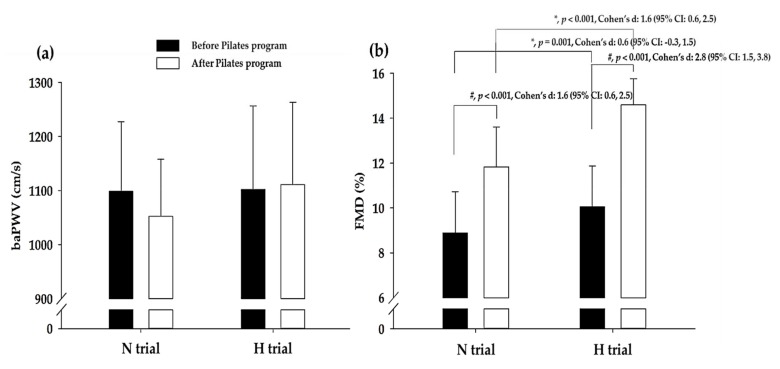
baPWV and FMD data before and after the Pilates program in the N trial and H trial. (**a**) Change in baPWV before and after the Pilates program in each trial. (**b**) Change in FMD before and after the Pilates program in each trial. N trial: Pilates program under normoxic condition, H trial: Pilates program under hypoxic condition, CI: Confidence interval, baPWV: Brachial ankle pulse wave velocity, FMD: Flow-mediated vasodilation. ^#^ Significant difference between before and after Pilates program, ^#^
*p* < 0.05. * Significant difference between the N trial and H trial, * *p* < 0.05.

**Table 1 ijerph-17-02584-t001:** Metabolic parameters during the Pilates program in the N trial and H trial.

Variables	N Trial	H Trial	Cohen’s *d* (95% CI)	*t*-Value	*p*-Value
VE (L/50 min)	1021.6 ± 196.7	1161.2 ± 188.7	0.7 (−0.2, 1.6)	−3.191	0.011 *
VO_2_ (mL/50 min)	32,299.7 ± 5581.0	32,910.5 ± 5407.3	0.1 (−0.7, 0.9)	−0.747	0.474
VCO_2_ (mL/50 min)	25,967.4 ± 4403.5	29,983.5 ± 4946.7	0.8 (−0.1, 1.7)	−4.597	0.001 *
RER	0.8 ± 0.0	0.9 ± 0.0	3.2 (1.8, 4.3)	−9.937	0.000 *
EE (kcal/50 min)	160.3 ± 27.6	168.3 ± 25.3	0.3 (−0.6, 1.1)	−1.746	0.115
CHO (g/50 min)	14.0 ± 4.2	29.1 ± 6.2	2.8 (1.5, 3.8)	−8.494	0.000 *
FAO (g/50 min)	10.6 ± 2.8	5.1 ± 2.0	−2.1 (−3.1, −1.0)	8.891	0.000 *

Values are expressed as means ± standard deviations, N trial: Pilates program under normoxic condition, H trial: Pilates program under hypoxic condition, CI: confidence interval, VE: Minute ventilation, VO_2_: Oxygen consumption, VCO_2_: Carbon dioxide production, RER: Resting energy requirement, EE: Energy expenditure, CHO: Carbohydrate oxidation, FAO: Fatty acid oxidation. * Significant difference between N trial and H trial, * *p* < 0.05.

**Table 2 ijerph-17-02584-t002:** Cardiac function parameters during the Pilates program in the N trial and H trial

Variables	N Trial	H Trial	Cohen’s *d* (95% CI)	*t*-Value	*p*-Value
HR (bpm/50 min)	5139.8 ± 656.2	5677.3 ± 553.9	0.9 (0.0, 1.7)	−3.204	0.011 *
SV (mL/50 min)	3848.9 ± 530.2	3618.9 ± 224.0	−0.5 (−1.3, 0.4)	1.538	0.158
CO (L/50 min)	395.7 ± 56.1	416.0 ± 45.6	0.4 (−0.5, 1.2)	−1.147	0.281
EDV (mL/50 min)	4954.9 ± 657.1	4624.6 ± 355.7	−0.6 (−1.4, 0.3)	1.750	0.114
ESV (mL/50 min)	1106.0 ± 268.6	995.2 ± 281.4	−0.4 (−1.2, 0.5)	1.453	0.180
EF	77.7 ± 4.7	78.7 ± 5.0	0.2 (−0.6, 1.0)	−0.818	0.435

Values are expressed as means ± standard deviations, N trial: Pilates program under normoxic condition, H trial: Pilates program under hypoxic condition, CI: confidence interval, HR: Heart rate, SV: Stroke volume, CO: Cardiac output, EDV: End-diastolic volume, ESV: End-systolic volume, EF: Ejection fraction. * Significant difference between N trial and H trial, * *p* < 0.05.

**Table 3 ijerph-17-02584-t003:** Blood pressure data before and after the Pilates program in the N trial and H trial

Variables	N Trial				H Trial				*F*-Value (*η*^2^)		
	Before	After	Cohen’s *d* (95% CI)	*p*-Value	Before	After	Cohen’s *d* (95% CI)	*p*-Value	Time	Group	Interaction
HR (bpm/min)	76.7 ± 7.5	102.8 ± 13.1	2.4 (1.2, 3.4)	0.000	86.9 ± 7.5	113.5 ± 11.1	2.8 (1.5, 3.8)	0.000	13.738 (0.604) ^†^	51.278 (0.851) ^†^	0.061 (0.007)
SBP (mmHg)	111.3 ± 8.4	109.1 ± 8.0	−0.3 (−1.1, 0.6)	0.343	108.4 ± 8.9	107.1 ± 7.3	−0.1 (−1.0, 0.7)	0.325	1.391 (0.134)	1.945 (0.178)	0.203 (0.022)
DBP (mmHg)	66.2 ± 8.4	62.1 ± 10.2	−0.4 (−1.3, 0.4)	0.126	66.4 ± 8.5	63.3 ± 8.5	−0.4 (−1.2, 0.5)	0.115	4.872 (0.351)	0.167 (0.018)	0.133 (0.015)
MAP (mmHg)	81.2 ± 7.5	77.3 ± 8.9	−0.5 (−1.3, 0.4)	0.089	80.7 ± 8.0	77.9 ± 7.3	−0.4 (−1.2, 0.5)	0.117	5.286 (0.370) ^†^	0.002 (0.000)	0.232 (0.025)
PP (mmHg)	45.1 ± 8.0	47.1 ± 5.6	0.3 (−0.6, 1.1)	0.321	42.1 ± 7.7	43.8 ± 7.6	0.2 (−0.6, 1.1)	0.309	1.683 (0.158)	4.293 (0.323)	0.022 (0.002)

Values are expressed as means ± standard deviations, N trial: Pilates program under normoxic condition, H trial: Pilates program under hypoxic condition, CI: confidence interval, SBP: Systolic blood pressure, DBP: Diastolic blood pressure, MAP: Mean arterial pressure, PP: Pulse pressure. †Significant interaction or main effect, ^†^
*p* < 0.05.

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
