# Peer review of "Effects of an Acute Pilates Program under Hypoxic Conditions on Vascular Endothelial Function in Pilates Participants: A Randomized Crossover Trial"

_ijerph, 2020, doi:10.3390/ijerph17072584_

Round 1

Reviewer 1 Report

The authors have carried out a Pilates training program under two conditions: normoxia vs hypoxia. The authors show very interesting results. However, I have a number of comments and suggestions that should be answered before considering the manuscript for publication.

Title: Please include in the title the type of study.

Introduction:

Page 1. Lines: 33-40. The following paragraph is not supported by the studies cited: "Pilates programs are relatively low intensity compared to endurance exercises that can induce high interest to sedentary/overweight/obese people, and they are also useful for strengthening the center of the body and increasing flexibility [3,4]". For example, the reference: de Oliveira Francisco, C.; de Almeida Fagundes, A.; Gorges, B. Effects of Pilates method in elderly people: Systematic review of randomized controlled trials. Journal of bodywork and movement therapies. 2015, 19, 500-508, doi:10.1016/j.jbmt.2015.03.003. It does not compare the effects of Pilates with endurance training.

Page 1. Lines: 60-67. I think the authors have not justified the problem of the study and the use of the pilates training combination under hypoxic conditions.   The authors should work deeper into why it would be beneficial to conduct Pilates sessions under hypoxic conditions.

Materials and methods:

The methodology has been adequately described. However, I have some suggestions. 

The training sessions were randomized?. I have noticed that the authors only explain it in the figure, but not in the text. Please describe how this process was carried out.

Did participants acclimate to the condition of hypoxia before conducting the session?

In statistical analysis, the authors should explain what assumptions were evaluated. Before applying an ANOVA, the assumptions should be checked and the authors describe only the test of normality. In addition, the statistical analysis performed is not clear. Why two ways, the authors have only one group evaluated several times. This type of objective is evaluated with repeated measures ANOVA. For those sets of variables that have been evaluated during the sessions should calculate the size of the effect with dCohen. The authors should provide a table for each set of variables showing the ANOVA values (F value of the main effect, the degrees of freedom, the p value and the effect size). In addition, as the authors present the research problem, I suggest that for each set of variables a MANOVA be applied.

Results:

I suggest that the authors present a descriptive table with the participants' baseline values. In addition, authors should not repeat the values. The values shown in the table should not be described in the text. In Table 1 and 2, should show the p value obtained in the t-test. Please, also show the p-values in table 3

Discussion

Page 7. Lines: 245-270. Authors should include the following study:

 doi: 10.1007/s00421-017-3571-7

Author Response

Point-by-Point Responses to the Reviewers

We thank the reviewers for their guidance for further improving our revised manuscript (ijerph-756212) entitled “Synergistic effects of an acute Pilates program in hypoxic conditions on vascular endothelial function in Pilates participants”. As described below, we have responded to all the comments brought up by the reviewers and incorporated all the changes suggested by the reviewer. In addition, we have received English corrections from the English grammar editing company, Editage.

Reviewer 1.

Review Report Form

Comments and Suggestions for Authors

The authors have carried out a Pilates training program under two conditions: normoxia vs hypoxia. The authors show very interesting results. However, I have a number of comments and suggestions that should be answered before considering the manuscript for publication.

  1. Title: Please include in the title the type of study.

- We included the study type in the title as follows: Synergistic effects of an acute Pilates program in hypoxic conditions on vascular endothelial function in Pilates participants: A randomized crossover trial.

Introduction:

  1. Page 1. Lines: 33-40. The following paragraph is not supported by the studies cited: "Pilates programs are relatively low intensity compared to endurance exercises that can induce high interest to sedentary/overweight/obese people, and they are also useful for strengthening the center of the body and increasing flexibility [3,4]". For example, the reference: de Oliveira Francisco, C.; de Almeida Fagundes, A.; Gorges, B. Effects of Pilates method in elderly people: Systematic review of randomized controlled trials. Journal of bodywork and movement therapies. 2015, 19, 500-508, doi:10.1016/j.jbmt.2015.03.003. It does not compare the effects of Pilates with endurance training.

: Thanks for the reviewer 1 comments. We agree with your comments. Therefore, the citation of the reference has been changed as follows: In particular, Pilates programs using tubing bands has the advantage of minimizing the possibility of injury to joints and muscles, and freely adjusting the intensity according to one's own muscle strength and physical fitness [2,3]. Pilates programs are relatively low intensity compared to endurance exercises that can induce high interest to sedentary/overweight/obese people, and they are also useful for strengthening the center of the body and increasing flexibility [4,5].

Reference

  1. Patterson, R.M.; Stegink Jansen, C.W.; Hogan, H.A.; Nassif, M.D. Material properties of Thera-Band Tubing. Physical therapy 2001, 81, 1437-1445, doi:10.1093/ptj/81.8.1437.
  2. Åžavkin, R.; Aslan, UB. The effect of Pilates exercise on body composition in sedentary overweight and obese women. The Journal of Sports Medicine and Physical Fitness 2017, 57, 1464-1470. doi: 10.23736/S0022-4707.16.06465-3.
  3. Rayes, ABR.; de Lira, CAB; Viana, RB; Benedito-Silva, AA; Vancini, RL; Mascarin, N; Andrade, MS. The effects of Pilates vs. aerobic training on cardiorespiratory fitness, isokinetic muscular strength, body composition, and functional tasks outcomes for individuals who are overweight/obese: a clinical trial. PeerJ 2019, 7, e6022. doi: 10.7717/peerj.6022.
  4. Jago, R.; Jonker, M.L.; Missaghian, M.; Baranowski, T. Effect of 4 weeks of Pilates on the body composition of young girls. Preventive medicine 2006, 42, 177-180, doi:10.1016/j.ypmed.2005.11.010.

  1. Page 1. Lines: 60-67. I think the authors have not justified the problem of the study and the use of the pilates training combination under hypoxic conditions. The authors should work deeper into why it would be beneficial to conduct Pilates sessions under hypoxic conditions.

: Thanks for the good comments. We revised the contents of Lines 60-67 on page 1 as follows: As mentioned earlier, Pilates program is not effective in enhancing cardiovascular function due to the lower exercise intensity. Therefore, to maximize the effect of Pilates program, it is very important to examine whether the combination of Pilates and hypoxic condition has additional effects on cardiac and vascular function. In other words, the combination of Pilates program and hypoxic conditions is likely to be an effective method to improve musculoskeletal function and cardiovascular function while reducing the risk of injury by reducing exercise load and mechanical stress. Although the exercise type is different, many previous studies have used exercise intervention in hypoxic conditions to show a greater improvement in body composition, metabolic function, and cardiovascular function with lower exercise intensity than exercise interventions in normoxic conditions [12,13,17,24,26,27]. Therefore, it is very important to examine the synergistic effects of Pilates programs and hypoxic conditions in improving metabolic, cardiac, and vascular functions.

Materials and methods:

The methodology has been adequately described. However, I have some suggestions.

  1. The training sessions were randomized?. I have noticed that the authors only explain it in the figure, but not in the text. Please describe how this process was carried out.

: We wrote that the study was conducted in a randomized crossover experimental trial in the study design section. The contents are as follows: All subjects visited the laboratory three times during the experimental period. During the first visit, they performed a familiarization trial under normoxic conditions (inspired oxygen fraction; FiO2 = 20.9%) prior to the main trial for adaptation to the Pilates program. On the second and third occasions, subjects carried out a randomized crossover experimental trial under either hypoxic conditions (FiO2 = 14.5%, a 3000-m simulated altitude; H trial) or normoxic conditions (N trial). All subjects had a washout period of at least 7-days between each trial.

  1. Did participants acclimate to the condition of hypoxia before conducting the session?

: In the present study, all trials was performed in a 9-m (width) × 7-m (length) × 3-m (high) chamber with a temperature of 23 ± 1°C and humidity of 50% ± 5% regulated by an environmental control chamber (NCTC-1, Nara control, Seoul, Korea). Therefore, all subjects were exposed to each environmental condition for 30-min. Therefore, we wrote the following in the text: During all trials, all subjects were exposed to each environmental condition for 30-min, and then a tubing Pilates program was performed for 50-min in each environmental condition.

  1. In statistical analysis, the authors should explain what assumptions were evaluated. Before applying an ANOVA, the assumptions should be checked and the authors describe only the test of normality. In addition, the statistical analysis performed is not clear. Why two ways, the authors have only one group evaluated several times. This type of objective is evaluated with repeated measures ANOVA. For those sets of variables that have been evaluated during the sessions should calculate the size of the effect with Cohen’s d. The authors should provide a table for each set of variables showing the ANOVA values (F value of the main effect, the degrees of freedom, the p value and the effect size). In addition, as the authors present the research problem, I suggest that for each set of variables a MANOVA be applied.

: Thank you for the suggestion. The statistical analysis has been revised.

“The assumption of normality and homoscedasticity was verified using the Sharpiro-Wilks W-test prior to using the parametric tests. First, independent t-test was analyzed to compare metabolic function and cardiac function difference between N-trial and H-trial Pilates program. Second, two-way analysis of variance (ANOVA) with repeated measures was used to assess the presence of interactions (trial × time) and main effects (trial, time) in vascular function before and after the Pilates program in each environmental condition. When ANOVA revealed a significant interaction or main effect, the Bonferroni test was performed as a post-hoc analysis to identify the differences. Additionally, the paired t-test was used to compare the metabolic and cardiac function during exercise between the N-trial and H-trial Pilates programs. p < 0.05 was considered to indicate statistical significance.”

Results:

  1. I suggest that the authors present a descriptive table with the participants' baseline values. In addition, authors should not repeat the values. The values shown in the table should not be described in the text. In Table 1 and 2, should show the p value obtained in the t-test. Please, also show the p-values in table 3

: Thank you for the suggestion. The baseline value of the subject is written in the text. Also, as suggested by the reviewer has been added and modified as:

“The study subjects were ten healthy females who were experienced in Pilates (age, 26.4 ± 3.0 yrs; height, 162.2 ± 4.1 cm; weight, 50.8 ± 5.8 kg; body mass index, 19.3 ± 1.6; fat free mass, 38.2 ± 3.8 kg; percent of body fat, 24.6% ± 5.0%),”

Discussion

  1. Page 7. Lines: 245-270. Authors should include the following study:

doi: 10.1007/s00421-017-3571-7

: As reviewer 1’s comments, We included the following study in the discussion section: Ramos-Campo DJ, Rubio-Arias JA, Dufour S, Chung L, Ávila-Gandía V, Alcaraz PE. Biochemical responses and physical performance during high-intensity resistance circuit training in hypoxia and normoxia. Eur J Appl Physiol. 117(4):809-818. doi: 10.1007/s00421-017-3571-7. Epub 2017 Mar 4. The contents are as follows: Additionally, exercise in hypoxic conditions results in a decrease in oxygen partial pressure of the arterial blood, which leads to metabolic acidosis through increase of ATP synthesis by the anaerobic metabolic process, increase of hydrogen ion, and decrease of pH [36]. Ramos-Campo et al. [37] was to verify the effect of various hypoxic condition on metabolic and acid-base balance, blood oxygenation, electrolyte, and half-squat performance parameters during high-resistance circuit (HRC) training. They reported that various normobaric hypoxic condition during HRC exercise reduce blood oxygenation, pH, and HCO3-, and increased blood lactate ultimately decreasing muscular performance.

Reviewer 2 Report

This is a study examining the influence of exercise (Pilates) in normoxic vs hypoxic conditions on metabolic and hemodynamic measures and will add new information.

The introduction provides sufficient background information and methods employed are adequate to adress the research question. Overall the study is methodologically sound.

While hypoxic condition had no influence on cardiac function, FMD was increased, and metabolism relied more on carbohydrate and less on fat oxidation.

I do not agree with the authors that the heart rate difference of the two conditions (111 vs 113 bpm) is of any physiological relevance, taking into account that all other CV measures did not differ. A detailed statement is desirable.

Taking age and heart rate of subjects into accound Pilates seems to be a rather moderate, aerobic exercise. The authors should address this issue.

The interpretation of data is complicated by the quite uncommon presentation which requires a calculater. Thus, I suggest to present data in the tables 1+2 in the usual way (per minute) which will make it easier for the reader.

Furthermore the inclusion of heart rate in Table 3 would be very welcome.

The discussion reads a bit dense with disjointed information and would benefit from re-organization.

Author Response

Point-by-Point Responses to the Reviewers

We thank the reviewers for their guidance for further improving our revised manuscript (ijerph-731013) entitled “Interval hypoxic training enhances athletic performance and does not adversely affect immune function in middle- and long-distance runners”. As described below, we have responded to all the comments brought up by the reviewers and incorporated all the changes suggested by the reviewer. In addition, we have received English corrections from the English grammer editing company, Editage.

Reviewer 2.

Review Report Form

Comments and Suggestions for Authors

This is a study examining the influence of exercise (Pilates) in normoxic vs hypoxic conditions on metabolic and hemodynamic measures and will add new information.

The introduction provides sufficient background information and methods employed are adequate to address the research question. Overall the study is methodologically sound.

While hypoxic condition had no influence on cardiac function, FMD was increased, and metabolism relied more on carbohydrate and less on fat oxidation.

  1. I do not agree with the authors that the heart rate difference of the two conditions (111 vs 113 bpm) is of any physiological relevance, taking into account that all other CV measures did not differ. A detailed statement is desirable.

: The heart rate at rest and the average heart rate during exercise in both trials are shown in the table. As shown in the table, there is a statistically significant difference in heart rate between the two trials. It is questionable where the sentence "the heart rate difference of the two conditions (111 vs 113 bpm)" mentioned above comes from.

Variables

N trial

H trial

Mean change (95% CI)

t-value

HR_rest (bpm/min)

76.7±7.5

86.9±7.5

10.2 (4.1, 16.3)

-3.774**

HR_Ex (bpm/min)

102.8±13.1

113.5±11.1

10.8 (3.2, 18.3)

-3.204*

  1. Taking age and heart rate of subjects into account Pilates seems to be a rather moderate, aerobic exercise. The authors should address this issue.

: Pilates presents various exercise intensities (from light to moderate) depending on the program composition. The Pilates program used in this study is a method that is generally applied to sedentary / overweight / obese people in the field. The heart rate difference between N trial and H trial is as follows: 102.8 ± 13.1 bpm (N trial) vs 113.5 ± 11.1 bpm (H trial). This heart rate corresponds to a light exercise. Therefore, we tried to confirm the synergy effect by combining this Pilates program in this study with hypoxic condition.

  1. The interpretation of data is complicated by the quite uncommon presentation which requires a calculater. Thus, I suggest to present data in the tables 1+2 in the usual way (per minute) which will make it easier for the reader.

: In the present study, Table 1 and 2 show the difference in metabolic and cardiac functions between trials during a typical 50-min Pilates program. It is more advantageous to check summation data to verify clear differences in metabolic and cardiac function during the Pilates program [Park et al., 2018;2019; Jung et al., 2020]. The data per minute can be checked by dividing the summation value by 50. Again, we believe that the summation value has a greater physiological value in interpretation than the data per minute. If reviewer 2 comments you must check the data per minute, we will provide you with data on this as a supplementation file.

<Reference>

Park, H.Y.; Park, W.; Lim, K. Living High-Training Low for 21 Days Enhances Exercise Economy, Hemodynamic Function, and Exercise Performance of Competitive Runners. Journal of sports science & medicine 2019, 18, 427-437.

Park, H.Y.; Shin, C.; Lim, K. Intermittent hypoxic training for 6 weeks in 3000 m hypobaric hypoxia conditions enhances exercise economy and aerobic exercise performance in moderately trained swimmers. Biology of sport 2018, 35, 49-56, doi:10.5114/biolsport.2018.70751.

Jung, W.S.; Kim, S.W.; Park, H.Y. Interval Hypoxic Training Enhances Athletic Performance and Does Not Adversely Affect Immune Function in Middle- and Long-Distance Runners. Int J Environ Res Public Health. 2020, 17(6), pii: E1934. doi: 10.3390/ijerph17061934.

  1. Furthermore the inclusion of heart rate in Table 3 would be very welcome.

: As reviewer 2 comments, we added heart rate data at rest to Table 3.

  1. The discussion reads a bit dense with disjointed information and would benefit from re-organization.

: In the discussion section, we prepared a discussion based on the results of previous studies and physiological mechanisms in accordance with the content presented in the results. As reviewer 2 comments, we wrote the discussion in the order of metabolic function, cardiac function, and vascular function for readers' convenience. However, it is judged that it is unreasonable to divide each section because there are overlapping explanations between the independent variables metabolic function, cardiac function, and vascular function. We appreciate your understanding of this.

Reviewer 3 Report

GENERAL COMMENTS

a) The manuscript needs to be proofread by someone who writes idiomatic English.  The current version contains too many grammar errors (mostly mismatched nouns and verbs, singular/plural issues).

b) The discussion is much too long.

c) The repeated claims of priority ("..is the first to combine.." are annoying. 

d) The manuscript repeatedly refers to "synergistic effects" but fails to indicate which quantitative methods were used to determine that the combination effect was indeed greater than what would have been expected from the individual effects.

DATA PRESENTATION

Table 1:  The legend defines a single asterisk (*) as both p<0.05 and p<0.001.

Table 2:  Again, the legend defines a single asterisk as both p<0.05 and p<0.001.  Furthermore, it defines two ** as p<0.01, yet the table shows no data sets with a p-value other than 0.05.

Table 3:  The legend defines the symbols *, **, and ***, and †,††, and ††† to indicate p-values of <0.05, <0.01, and <0.001, respectively, but no differences between data sets shown in the table are identified as significantly different.

Figure 1:  The word "performed" serves no useful purpose and should be deleted.

Figure 2:  Again, the legend defines p-values that do not occur in the figure, and a single asterisk (*) is used to define p-values of both p<0.05 and p<0.001.

Lines 98-103:  Some readers might not be familiar with these short descriptions of Pilates exercises.  A reference to a publications or a website would be helpful.

Author Response

Point-by-Point Responses to the Reviewers

We thank the reviewers for their guidance for further improving our revised manuscript (ijerph-731013) entitled “Interval hypoxic training enhances athletic performance and does not adversely affect immune function in middle- and long-distance runners”. As described below, we have responded to all the comments brought up by the reviewers and incorporated all the changes suggested by the reviewer. In addition, we have received English corrections from the English grammer editing company, Editage.

Reviewer 3.

Review Report Form

GENERAL COMMENTS

  1. a) The manuscript needs to be proofread by someone who writes idiomatic English. The current version contains too many grammar errors (mostly mismatched nouns and verbs, singular/plural issues).

: We have received English corrections from the English grammer editing company, Editage. We have attached an English proofreading certificate. Nevertheless, if additional English correction is required, we will take the English correction again in the next review round.

  1. b) The discussion is much too long.

: In the discussion section, we prepared a discussion based on the results of previous studies and physiological mechanisms in accordance with the content presented in the results. As reviewer 3 comment, the discussion is too long in quantity. However, we've made every effort to include all comments from 3 reviewers and to faithfully interpret the resulting data. We appreciate your understanding of this.

  1. c) The repeated claims of priority ("..is the first to combine.." are annoying.

: We have made corrections to this. The revised sentence is as follows: However, the present study is the new insight to combine a Pilates program and hypoxic conditions, and it is necessary to examine the differences in the metabolic, cardiac, and vascular function responses under hypoxic and normoxic conditions in Pilates participants who can perform Pilates programs with accurate movements.

  1. d) The manuscript repeatedly refers to "synergistic effects" but fails to indicate which quantitative methods were used to determine that the combination effect was indeed greater than what would have been expected from the individual effects.

: The "synergistic effects" we're talking about here are talking about the difference in effectiveness for the two trials (N trial vs H trial). And our study confirmed that a Pilates program in a moderately hypoxic condition showed a synergistic effect on vascular endothelial function before and after the Pilates program in hypoxic compared to normoxic conditions in experienced Pilates participants. The H trial showed a statistically greater %FMD improvement than the N trial. If reviewer 3 tell us exactly what the problem is, we will correct it accordingly.

DATA PRESENTATION

Table 1: The legend defines a single asterisk (*) as both p<0.05 and p<0.001.

: Thank you very much for your comments. As suggested by the reviewer has been deleted and modified.

Table 2: Again, the legend defines a single asterisk as both p<0.05 and p<0.001. Furthermore, it defines two ** as p<0.01, yet the table shows no data sets with a p-value other than 0.05.

: Thank you very much for your comments. As suggested by the reviewer has been deleted and modified.

Table 3: The legend defines the symbols *, **, and ***, and †,††, and ††† to indicate p-values of <0.05, <0.01, and <0.001, respectively, but no differences between data sets shown in the table are identified as significantly different.

: Thank you very much for your comments. As suggested by the reviewer has been deleted and modified.

Figure 1: The word "performed" serves no useful purpose and should be deleted.

: As reviewer 3 comments, we deleted the word "performed".

Figure 2: Again, the legend defines p-values that do not occur in the figure, and a single asterisk (*) is used to define p-values of both p<0.05 and p<0.001.

: Thank you very much for your comments. As suggested by the reviewer has been deleted and modified.

Lines 98-103: Some readers might not be familiar with these short descriptions of Pilates exercises. A reference to a publications or a website would be helpful.

: Thank you very much for your comments. As suggested by the reviewer has been added and modified.

Round 2

Reviewer 1 Report

Thank you very much for the answers. I think the manuscript has improved in quality. However, I believe that some of the comments I had proposed have not yet resulted. I will now describe them.

Regarding this comment

The training sessions were randomized?. I have noticed that the authors only explain it in the figure, but not in the text. Please describe how this process was carried out.

: We wrote that the study was conducted in a randomized crossover experimental trial in the study design section. The contents are as follows: All subjects visited the laboratory three times during the experimental period. During the first visit, they performed a familiarization trial under normoxic conditions (inspired oxygen fraction; FiO2 = 20.9%) prior to the main trial for adaptation to the Pilates program. On the second and third occasions, subjects carried out a randomized crossover experimental trial under either hypoxic conditions (FiO2 = 14.5%, a 3000-m simulated altitude; H trial) or normoxic conditions (N trial). All subjects had a washout period of at least 7-days between each trial.

The authors should explain the process of randomization and also whether the experimental process was blinded

“The assumption of normality and homoscedasticity was verified using the Sharpiro-Wilks W-test prior to using the parametric tests”.

I agree.

First, independent t-test was analyzed to compare metabolic function and cardiac function difference between N-trial and H-trial Pilates program.

I completely disagree. The authors should have applied a paired sample t-test. It's the same participants exposed to different scenarios.

Second, two-way analysis of variance (ANOVA) with repeated measures was used to assess the presence of interactions (trial × time) and main effects (trial, time) in vascular function before and after the Pilates program in each environmental condition.

I still don't know how the authors calculated the effect on trial and time. There are 4 evaluation moments in the same subjects. Please explain to me where you get/calculate the trial and time

When ANOVA revealed a significant interaction or main effect, the Bonferroni test was performed as a post-hoc analysis to identify the differences.

I agree.

Additionally, the paired t-test was used to compare the metabolic and cardiac function during exercise between the N-trial and H-trial Pilates programs.

what is the difference from the first t-test?

p < 0.05 was considered to indicate statistical significance.”

I agree.

Thank you for the suggestion. The baseline value of the subject is written in the text. Also, as suggested by the reviewer has been added and modified as: “The study subjects were ten healthy females who were experienced in Pilates (age, 26.4 ± 3.0 yrs; height, 162.2 ± 4.1 cm; weight, 50.8 ± 5.8 kg; body mass index, 19.3 ± 1.6; fat free mass, 38.2 ± 3.8 kg; percent of body fat, 24.6% ± 5.0%)”

Thank you for following my recommendations. However, in the methodology section the authors should explain the sampling process and how many were recruited and in the results section the final sample that completed the study and its characteristics (CONSORT recommendations)

The authors should provide a table for each set of variables showing the ANOVA values (F value of the main effect, the degrees of freedom, the p value and the effect size).

It has not been reviewed.

On the other hand, the authors have not shown the p-value for each difference (t-test) and have not calculated the effect size. Furthermore, the authors continue to repeat the information in tables and text.

Author Response

Point-by-Point Responses to the Reviewers

We thank the reviewers for their guidance for further improving our revised manuscript (ijerph-756212) entitled “Synergistic effects of an acute Pilates program in hypoxic conditions on vascular endothelial function in Pilates participants”. As described below, we have responded to all the comments brought up by the reviewers and incorporated all the changes suggested by the reviewer. In addition, we have received additional English corrections from the English grammar editing company, Editage.

Reviewer 1.

Review Report Form

Comments and Suggestions for Authors

Thank you very much for the answers. I think the manuscript has improved in quality. However, I believe that some of the comments I had proposed have not yet resulted. I will now describe them.

Regarding this comment

The training sessions were randomized?. I have noticed that the authors only explain it in the figure, but not in the text. Please describe how this process was carried out.

: We wrote that the study was conducted in a randomized crossover experimental trial in the study design section. The contents are as follows: All subjects visited the laboratory three times during the experimental period. During the first visit, they performed a familiarization trial under normoxic conditions (inspired oxygen fraction; FiO2 = 20.9%) prior to the main trial for adaptation to the Pilates program. On the second and third occasions, subjects carried out a randomized crossover experimental trial under either hypoxic conditions (FiO2 = 14.5%, a 3000-m simulated altitude; H trial) or normoxic conditions (N trial). All subjects had a washout period of at least 7-days between each trial.

  1. The authors should explain the process of randomization and also whether the experimental process was blinded.

: Thanks for the good comment. As reviewer 1’s comment, we added the following: The order of the conditions for each experimental trial was randomized, and each subject performed the protocol at the same time of date for each visit. However, subjects did not conduct blinded experiments on environmental conditions. This is a limitation of this study.

“The assumption of normality and homoscedasticity was verified using the Shapiro-Wilks W-test prior to using the parametric tests”.

I agree.

First, independent t-test was analyzed to compare metabolic function and cardiac function difference between N-trial and H-trial Pilates program.

  1. I completely disagree. The authors should have applied a paired sample t-test. It's the same participants exposed to different scenarios.

: This is our obvious mistake. We conducted a paired t-test. Therefore, the sentence was modified as follows: First, paired t-test was analyzed to compare metabolic function and cardiac function difference between N-trial and H-trial Pilates program.

Second, two-way analysis of variance (ANOVA) with repeated measures was used to assess the presence of interactions (trial × time) and main effects (trial, time) in vascular function before and after the Pilates program in each environmental condition.

  1. I still don't know how the authors calculated the effect on trial and time. There are 4 evaluation moments in the same subjects. Please explain to me where you get/calculate the trial and time.

: There were a few mistakes we made in writing statistical methods. We apologize for this. First, we performed a normality and homoscedasticity test using the Shapiro-Wilks W-test prior to using the parametric tests. And then, paired t-test was analyzed to compare metabolic function and cardiac function difference between N-trial and H-trial Pilates program. Second, two-way analysis of variance (ANOVA) with repeated measures was performed to verify the interaction effect according to environmental condition (N trial or H trial) and time (before or after) on vascular function. In order to strictly apply the repeated two ANOVA, we conducted a post analysis only when the interaction effect appeared. When ANOVA revealed a significant interaction, the Bonferroni test was performed as a post-hoc analysis to identify the differences (4 evaluation: difference between before and after Pilates program in N trial and H trial, difference between N trial and H trial in before and after Pilates program).

Especially, as shown in Table 3, post-hoc analysis was not performed for heart rate with no interaction effect. Therefore, the significance of p-value (*) was not indicated in N trial and H trial.

When ANOVA revealed a significant interaction or main effect, the Bonferroni test was performed as a post-hoc analysis to identify the differences.

I agree.

Additionally, the paired t-test was used to compare the metabolic and cardiac function during exercise between the N-trial and H-trial Pilates programs.

  1. what is the difference from the first t-test?

: This is our obvious mistake. We have removed this.

p < 0.05 was considered to indicate statistical significance.”

I agree.

Thank you for the suggestion. The baseline value of the subject is written in the text. Also, as suggested by the reviewer has been added and modified as: “The study subjects were ten healthy females who were experienced in Pilates (age, 26.4 ± 3.0 yrs; height, 162.2 ± 4.1 cm; weight, 50.8 ± 5.8 kg; body mass index, 19.3 ± 1.6; fat free mass, 38.2 ± 3.8 kg; percent of body fat, 24.6% ± 5.0%)”

  1. Thank you for following my recommendations. However, in the methodology section the authors should explain the sampling process and how many were recruited and in the results section the final sample that completed the study and its characteristics (CONSORT recommendations)

: Thanks for the good comment. As reviewer 1’s comment, we added the CONSORT flow diagram. Please check Figure 1.

The authors should provide a table for each set of variables showing the ANOVA values (F value of the main effect, the degrees of freedom, the p value and the effect size).

  1. It has not been reviewed. On the other hand, the authors have not shown the p-value for each difference (t-test) and have not calculated the effect size. Furthermore, the authors continue to repeat the information in tables and text.

: We apologize again for this. First, the mean change (95% CI) was changed to the effect size (95% CI) in Tables 1, 2, and 3, and the p value was added. Second, the F value for the interaction effect and the main effect is currently displayed in Table 3, and the degree of freedom corresponds to n-1, so we do not think it is necessary to write. Also, this was also done in Figure 2. Eta-squared was used as the effect size for ANOVA, and cohen's d was used as the effect size for paired t-test.

Reviewer 3 Report

This revised version represents a modest improvement over the original manuscript.  A few deficiencies (such as some errors in the table legends) have been corrected.  Several issues have yet to be addressed in a satisfactory manner.

Grammar and Spelling:  The manuscript contains too many grammar and spelling errors.  In response to my previous review, the authors supplied a certificate that confirms that the manuscript has been edited by Editage, a commercial editing company.  The certificate is dated March11, 2020 and thus predates my review by about 9 days.  In other words, Editage did not get involved in response to my review.  As far as I can tell, every error I marked in the original manuscript is also present in the revised version.  The authors have made no corrections whatsoever.  To give a few typical examples:

Line 33: "alisthenics" should be changed to "calisthenics" (spelling)

Line 35: "Pilates programs....has the advantage.." should be changed to "Pilates programs ....have the advantage..." (plural)

Line 52: "Hypoxic conditions elicits.." should be changed to "Hypoxic conditions elicit..." (plural)

Figure 1:  "A randomized crossover trials" should be changed to "A randomized crossover trial" (singular).

Line 168: "Sharpiro-Wilks" should be changed to "Shapiro-Wilks" (spelling)

References:  The references do not follow a consistent format.  Some titles are capitalized; others are in lower case.  Few if any journal names are in the required (abbreviated and capitalized) format.

Data Presentation:  The authors like to use the term "synergistic".  How did they prove that the observed interaction was indeed synergistic as opposed to merely additive?  Which quantitative method did they use to establish synergy?

The Discussion is too long.

Author Response

Point-by-Point Responses to the Reviewers

We thank the reviewers for their guidance for further improving our revised manuscript (ijerph-756212) entitled “Synergistic effects of an acute Pilates program in hypoxic conditions on vascular endothelial function in Pilates participants”. As described below, we have responded to all the comments brought up by the reviewers and incorporated all the changes suggested by the reviewer. In addition, we have received additional English corrections from the English grammar editing company, Editage.

Reviewer 3.

Review Report Form

Comments and Suggestions for Authors

This revised version represents a modest improvement over the original manuscript. A few deficiencies (such as some errors in the table legends) have been corrected. Several issues have yet to be addressed in a satisfactory manner.

  1. Grammar and Spelling: The manuscript contains too many grammar and spelling errors. In response to my previous review, the authors supplied a certificate that confirms that the manuscript has been edited by Editage, a commercial editing company. The certificate is dated March11, 2020 and thus predates my review by about 9 days. In other words, Editage did not get involved in response to my review. As far as I can tell, every error I marked in the original manuscript is also present in the revised version. The authors have made no corrections whatsoever. To give a few typical examples:

Line 33: "alisthenics" should be changed to "calisthenics" (spelling)

Line 35: "Pilates programs....has the advantage.." should be changed to "Pilates programs ....have the advantage..." (plural)

Line 52: "Hypoxic conditions elicits.." should be changed to "Hypoxic conditions elicit..." (plural)

Figure 1: "A randomized crossover trials" should be changed to "A randomized crossover trial" (singular).

Line 168: "Sharpiro-Wilks" should be changed to "Shapiro-Wilks" (spelling)

: Thank you for the meticulous and kind comments. As reviewer 3 comments, we corrected all the wrong expressions. Also, the manuscript was revised according to reviewer comment, and we once again got proofreading from the English grammar editing company Editage. Thank you again for your attention to spelling and Grammar.

  1. References: The references do not follow a consistent format. Some titles are capitalized; others are in lower case. Few if any journal names are in the required (abbreviated and capitalized) format.

: Thank you for good comment. We have modified the format of all references in accordance with the IJERPH format.

  1. Data Presentation: The authors like to use the term "synergistic". How did they prove that the observed interaction was indeed synergistic as opposed to merely additive? Which quantitative method did they use to establish synergy?

: We used the term “synergistic effect” only for the dependent variables with statistically interaction (trial â…¹ time effects) effects. In other words, we used the term “synergistic effect” for the dependent parameters (e.g. FMD) with a greater effect by the Pilates program under hypoxic condition vs normoxic condition. However, we removed all the term "synergistic" according to reviewer 3 comments.

  1. The Discussion is too long.

: Thanks for the kind comments. As reviewer 3 comments, we have reduced the discussion content as much as possible, considering what we intend to convey.

Round 3

Reviewer 1 Report

The authors have made the suggested changes. My latest comments are as follows: 

1. "Effects of an acute Pilates program": Please check the title, you cannot write acute and program. I would include the word session and not program. When you include the word program, it refers to chronic effects 

The authors have responded adequately in the letter, however, they have not made the same changes to the document. (statistics section).